# A survey of public attitudes toward uterus transplantation, surrogacy, and adoption in Japan

Akari Nakazawa[1], Tetsuya Hirata[1,2]*, Tomoko Arakawa[1], Natsuki Nagashima[1], Shinya Fukuda[1], Kazuaki Neriishi[1], Miyuki Harada[1], Yasushi Hirota[1], Kaori Koga[1], Osamu Wada-Hiraike[1], Yoshio Koizumi[2], Tomoyuki Fujii[1], Minoru Irahara[3], Yutaka Osuga[1]

1 Department of Obstetrics and Gynecology, Faculty of Medicine, University of Tokyo, Hongo, Bunkyo-ku, Tokyo, Japan, 2 Department of Obstetrics and Gynecology, Doai Kinen Hospital, Tokyo, Japan, 3 Department of Obstetrics and Gynecology, Tokushima University Graduate School of Biomedical Sciences, Tokushima, Japan

* thira-tky@umin.ac.jp

**Data Availability Statement:** All relevant data are within the manuscript and its Supporting Information files.

## Abstract

This study aimed to evaluate the attitudes of male and female members of the public toward uterus transplantation (UTx), surrogacy, and adoption in Japan via a web-based survey. One thousand six hundred participants were recruited with equal segregation of age (20s, 30s, 40s, and 50s) and gender. We assessed the association between ethical view and gender, age, infertility, and the knowledge level of UTx, using a questionnaire. The findings were as follows. First, 36.5% and 31.0% of respondents agreed that UTx and gestational surrogacy should be approved, respectively. Second, the respondents would potentially choose to receive UTx (34.4%), gestational surrogacy (31.9%), and adoption (40.3%), if they or their partners experienced absolute uterine factor infertility. Third, 10.1%, 5.8%, and 14.3% of the respondents chose UTx, gestational surrogacy, and adoption as the most favorable option, respectively. Fourth, if their daughters suffered from absolute uterine factor infertility, 32.3% of female respondents might want to be donors, and 36.7% of male respondents might ask their wives to be donors. These data were affected by age, gender, infertility, or the knowledge level of UTx. UTx was a more acceptable option than gestational surrogacy and adoption. The effects of gender, age, infertility, and the level of knowledge of UTx are important in understanding the attitude toward UTx. On the other hand, there were concerns about the safety of UTx for recipients, donors, and babies. It is important to continue to understand public attitudes to inform the development and safety of UTx, which will enhance the discussion on the ethical consensus on UTx.

## Introduction

In 2014, the first live birth after uterus transplantation (UTx) from a live donor was achieved in Gothenburg, Sweden [1]. Thereafter, recent trials of the procedure using living donors in Sweden and the USA have resulted in more than 10 live births [2]. Furthermore, the first live

**Funding:** This work was supported by Grant from the Ministry of Education, Culture, Sports, Science and Technology Grant from the Japan Agency for Medical Research and Development.

**Competing interests:** The authors have declared that no competing interests exist.

birth after UTx from a deceased donor was recently reported by Ejzenberg et al. [3]. Currently, trials of UTx have begun in other countries [4]. In Japan, many animal experiments have been conducted in preparation for the realization of UTx [5, 6]. Accordingly, UTx could be a potentially feasible option for patients with absolute uterine factor infertility in the future, although it is still at the early clinical experimental stage.

At the same time, the introduction of UTx as a clinical procedure raises major ethical implications. This technique would allow women who do not have a uterus (or who have undergone hysterectomy due to malignant or benign uterine disease) to carry their own fetuses. Thus, UTx can be an alternative to gestational surrogacy and adoption, and can also affect the attitude of the public about them. In Japan, in particular, gestational surrogacy is not performed [7, 8], and UTx might be considered as an alternative.

While discussing whether UTx should be performed in Japan, it is important to evaluate the public opinion in Japan regarding UTx to understand the perceived safety and ethical challenges. In a previous survey on UTx conducted by Kisu et al., 40% of the respondents, who were all females of reproductive age, felt positively about UTx [9]. However, females older than 40 years and males should be included to reflect the attitude of the public as a whole. Particularly, women older than 40 years may be involved in UTx as living donors, and it is important to survey their attitudes toward their potential roles as donors. Therefore, we conducted a survey of the attitudes toward UTx among male and female members of the public aged 20–60 years.

## Materials and methods

The study protocol was approved by the ethics committee of the University of Tokyo Hospital. This study was carried out as a part of a research initiative of the Japan Agency for Medical Research and Development. We conducted a web-based survey to assess the attitude of the public toward UTx, surrogacy, and adoption as we previously reported [10]. The sampling frame was developed by an internet research company (NEO MARKETING INC., Tokyo, Japan), which sent a questionnaire through the website and collected the responses. Respondents were asked to read a summary page explaining the purpose and content of the questionnaire prior to starting the survey. They provided informed consent before answering the questionnaire. Respondents read the study description and then chose to participate or not to participate in the study. Respondents started to answer the questionnaire after they selected "Yes" to a question of whether to participate in the survey. We also explained that respondents can stop answering and withdraw all answers if they want to withdraw from our survey halfway. Reference materials (S1 Fig) were also included to enable respondents to better understand UTx, surrogacy, and adoption when responding to the questionnaires. The questionnaire also included items regarding age, gender, marital status, number of children, and the experience of infertility (S1 Document). In Japan, infertility is defined as unsuccessful attempted conception for 12 months or longer. The web-based questionnaire was available online from October 24 to 25, 2017. In this program, only completed questionnaires could be submitted. One thousand six hundred participants were recruited with an equal segregation of age (20s, 30s, 40s, and 50s) and sex. Because we assumed that uterus transplantation donors and recipients could be women aged 20 to under 60, this study targeted men and women between the ages of 20 and 60. We asked the internet research company to collect 200 samples for each group of men and women in their 20s, 30s, 40s, and 50s. The upper limit of each sample collection was determined to be 250 samples. When the upper limit was reached, the collection of samples was closed, and further samples were not obtained. For each group, 200 samples were selected randomly. Through this process, the internet research company

extracted 1600 samples with equal strata of age and gender. Finally, we received the responses of these 1600 participants from the internet research company and subsequently used these for the analysis.

## Statistical analysis

The statistical analyses were performed using JMP Pro version 14 (SAS Institute Inc., Cary, NC, USA). Categorical data were analyzed using the chi-square test and Fisher's exact test, as appropriate, and are presented as numbers and percentages. P values < 0.05 were considered statistically significant.

## Results

### Characteristics of respondents

We sent out questionnaires to 14,990 people; 3931 responded and 1938 submitted completed responses (response rate, 12.9%). We randomly extracted 1600 samples with equal segments of age and gender. The study thus consisted of a total of 1600 respondents with an equal number across each age group (20s, 30s, 40s, and 50s). Responses from equal numbers of men and women in each age group were obtained. Among the respondents, 47.7% were unmarried and 52.3% were married. In this study, we asked whether respondents have any experience of infertility. The subjects were those who had infertility even if they had not undergone fertility treatment or screening. In this study, of the 307 respondents with experience of infertility, 93 did not undergo infertility screening or treatments. The characteristics of the respondents are shown in Table 1 and S1 Table.

### Social acceptance

Regarding the social and ethical acceptance of UTx, 36.5% of participants thought it "should be approved," 17.0% thought it "should *not* be approved," and 46.5% were "indecisive" (Table 2). A higher proportion of participants aged 30–39 years thought it "should be approved" than that of participants aged 50–59 years. Among participants aged 50–59 years, more females than males felt that UTx "should *not* be approved". Regarding the social acceptance of gestational surrogacy, 31.0% of participants felt that it "should be approved", 19.9% felt that it "should *not* be approved," and 49.1% were "indecisive." A higher number of male and female participants aged 20–29 years or 30–39 years felt that gestational surrogacy "should be approved" than that among participants aged 40–49 years or 50–59 years. These results were similar to those obtained for UTx.

As shown in Table 3, the frequency of a "positive attitude" toward UTx or gestational surrogacy was higher among males or females with infertility than among those without infertility, and indecisive responses were also less frequent in this group. Furthermore, participants were more likely to think that UTx and gestational surrogacy should be socially approved when they knew more about UTx (Table 4). However, more male participants who were familiar with UTx believed that "UTx should *not* be approved" or "gestational surrogacy should *not* be approved".

The most common reasons why respondents thought that UTx should be approved were: ". . .because UTx will give hope to people with absolute uterine infertility" (67.1%), ". . .because there is a genetic linkage with the born child, and the recipient can be a birth parent" (66.3%), and ". . .the risks associated with pregnancy and parturition are borne by the recipient and not third parties such as surrogate mothers" (44.3%) (S2 Table). In addition, the most frequent reasons why respondents thought that UTx should *not* be approved were: ". . .the risk of UTx

**Table 1. The demographics data of respondents (N = 1600).**

| | Total | male | female |
|---|---|---|---|
| **Age** | | | |
| 20–29 | 400 (25.0%) | 200 (25.0%) | 200 (25.0%) |
| 30–39 | 400 (25.0%) | 200 (25.0%) | 200 (25.0%) |
| 40–49 | 400 (25.0%) | 200 (25.0%) | 200 (25.0%) |
| 50–59 | 400 (25.0%) | 200 (25.0%) | 200 (25.0%) |
| **Marital status** | | | |
| Unmarried | 763 (47.7%) | 443 (55.4%) | 320 (40.0%) |
| Married | 837 (52.3%) | 357 (44.6%) | 480 (60.0%) |
| **Number of children** | | | |
| 0 | 891 (55.7%) | 501 (62.6%) | 390 (48.8%) |
| 1 | 283 (17.7%) | 113 (14.1%) | 170 (21.3%) |
| 2 | 330 (20.6%) | 142 (17.8%) | 188 (23.5%) |
| more than 2 | 96 (6.0%) | 44 (5.5%) | 52 (6.5%) |
| **Experience of infertility** | | | |
| Yes | 307 (19.2%) | 114 (14.3%) | 193 (24.1%) |
| None | 1293 (80.8%) | 686 (85.8%) | 607 (75.9%) |
| **Experience of infertility screening or treatment (responders or their spouses)** | | | |
| None | 1358 (84.9%) | 702 (87.8%) | 656 (82.0%) |
| Screening for infertility | 191 (11.9%) | 76 (9.5%) | 115 (14.4%) |
| Timed intercourse | 159 (9.9%) | 60 (7.5%) | 99 (12.4%) |
| IUI (intrauterine insemination) | 82 (5.1%) | 33 (4.1%) | 49 (6.1%) |
| IVF-ET (In vitro fertilization—embryo transfer) | 47 (2.9%) | 21 (2.6%) | 26 (3.3%) |
| ICSI (intracytoplasmic sperm injection) | 33 (2.1%) | 11 (1.4%) | 22 (2.8%) |
| **Do you know that woman, who underwent UTx, conceived and had a live birth with cesarean section?** | | | |
| Know very well | 79 (4.9%) | 47 (5.9%) | 32 (4.0%) |
| Know well | 130 (8.1%) | 83 (10.4%) | 47 (5.9%) |
| Have heard of it, but do not know much. | 404 (25.3%) | 217 (27.1%) | 187 (23.4%) |
| Have never heard of it. | 987 (61.7%) | 453 (56.6%) | 534 (66.8%) |

surgery is high" (52.6%), ". . .because the recipient is at a high risk during pregnancy and birth" (38.2%), and ". . .because UTx does not necessarily lead to pregnancy and birth" (36.4%)

**Table 2. The comparison of public attitude towards UTx and gestational surrogacy among various age and gender.**

| sex | age | group | N | Utx | | | gestational surrogacy | | |
|---|---|---|---|---|---|---|---|---|---|
| | | | | should be approved | should *not* be approved | indecisive | should be approved | should *not* be approved | indecisive |
| | | total | 1600 | **36.5%** | **17.0%** | **46.5%** | **31.0%** | **19.9%** | **49.1%** |
| male | 20–29 | A | 200 | 39.50% | 19.0% | 41.5% | 36.0% [D,G,H] | 20.5% | 43.5% |
| | 30–39 | B | 200 | 41.0% [D,H] | 20.0% | 39.0% | 35.5% [G,H] | 18.0% | 46.5% |
| | 40–49 | C | 200 | 34.0% | 15.5% | 50.5% [B,F] | 30.0% | 17.0% | 53.0% |
| | 50–59 | D | 200 | 31.0% | 13.0% | 56.0% [A,B,F] | 26.0% | 22.5% | 51.5% |
| female | 20–29 | E | 200 | 39.0% | 14.0% | 47.0% | 34.5% [H] | 17.0% | 48.5% |
| | 30–39 | F | 200 | 45.0% [C,D,G,H] | 15.0% | 40.0% | 36.5% [D,G,H] | 16.5% | 47.0% |
| | 40–49 | G | 200 | 32.0% | 18.0% | 50.0% [B] | 25.5% | 21.5% | 53.0% |
| | 50–59 | H | 200 | 30.5% | 21.5% [D] | 48.0% | 24.0% | 26.0% [C,E,F] | 50.0% |

[A,B,C,D,E,F,G,H] denotes significantly high compared to group A,B,C,D,E,F,G, or H respectively (P<0.05).

**Table 3. The comparison of public attitude towards UTx and gestational surrogacy between respondents with or without infertility.**

| sex | infertility | group | N | Utx | | | gestational surrogacy | | |
|---|---|---|---|---|---|---|---|---|---|
| | | | | should be approved | should *not* be approved | indecisive | should be approved | should *not* be approved | indecisive |
| | | total | 1600 | 36.5% | 17.0% | 46.5% | 31.0% | 19.9% | 49.1% |
| male | + | A | 114 | 45.6% [B,D] | 21.9% | 32.5% | 39.5% [D] | 22.8% | 37.7% |
| | - | B | 686 | 34.8% | 16.0% | 49.1% [A,C] | 30.6% | 19.0% | 50.4% [A,C] |
| female | + | C | 193 | 44.6% [B,D] | 17.6% | 37.8% | 40.9% [B,D] | 17.6% | 41.5% |
| | - | D | 607 | 34.1% | 17.0% | 48.9% [A,C] | 26.7% | 21.1% | 52.2% [A,C] |

[A,B,C,D] denotes significantly high compared to group A,B,C, or D respectively (P<0.05).

(S3 Table). The reasons why respondents thought that gestational surrogacy should be approved were: "...because both the client and the surrogate mother agree" (68.3%), "...because it is an option for women with absolute uterine infertility" (66.3%), and "...because the client can receive a child without undergoing a difficult operation" (47.6%) (S4 Table). Furthermore, the reasons why respondents thought that gestational surrogacy should *not* be approved were "...because the surrogate mother is at high risk during pregnancy and parturition" (46.9%) and "... the parent-child relationship will be unnatural" (41.5%) (S5 Table).

## Who is an eligible recipient or donor of UTx?

Next, we asked the question "Who is an eligible recipient of UTx?" In response, "a woman who do not have a uterus congenitally," "a woman who has had her uterus removed due to malignant or benign disease," and "a woman who has lost uterine function after endometritis or surgery for miscarriage" were regarded as acceptable recipients by 52.9%, 53.4%, 46.8%, and 46.4% of respondents, respectively (Table 5). More females than males responded that these women were acceptable recipients.

Further, we asked the question "Who is an eligible donor of UTx?" In response, "mother," "sister," and "deceased donor" (brain death or cardiac death) were regarded as acceptable donors by 39.4%, 37.9%, and 27.6% of respondents, respectively (Table 6).

**Table 4. The comparison of public attitude towards UTx and gestational surrogacy by degree of understanding of UTx.**

| sex | response | group | N | UTx | | | Gestational surrogacy | | |
|---|---|---|---|---|---|---|---|---|---|
| | | | | should be approved | should *not* be approved | indecisive | should be approved | should *not* be approved | indecisive |
| | | total | 1600 | 36.5% | 17.0% | 46.5% | 31.0% | 19.9% | 49.1% |
| male | Know very well | A | 47 | 46.8% | 36.2% [C,D,G,H] | 17.0% | 46.8% [C,D,H] | 36.2% [C,D,F,G,H] | 17.0% |
| | Know well | B | 83 | 42.2% | 30.1% [C,D,G,H] | 27.7% | 42.2% [D,H] | 21.7% | 36.1% [A] |
| | Have heard of it, but do not know much. | C | 217 | 36.4% | 17.1% | 46.5% [A,B] | 30.0% | 21.7% | 48.8% [A,E] |
| | Have never heard of it. | D | 453 | 34.2% | 12.4% | 53.4% [A,B,E,G] | 29.4% | 16.3% | 54.3% [A,B,E] |
| female | Know very well | E | 32 | 53.1% [D,H] | 18.8% | 28.1% | 56.3% [C,D,G,H] | 25.0% | 18.8% |
| | Know well | F | 47 | 42.6% | 19.1% | 38.3% [A] | 44.7% [D,H] | 12.8% | 42.6% [A,E] |
| | Have heard of it, but do not know much. | G | 187 | 39.0% | 17.6% | 43.3% [A,B] | 34.8% [H] | 18.2% | 47.1% [A,E] |
| | Have never heard of it. | H | 534 | 34.3% | 16.7% | 49.1% [A,B,E] | 25.7% | 21.3% | 53.0% [A,B,E] |

[A,B,C,D,E,F,G,H] denotes significantly high compared to group A,B,C,D,E,F,G, or H respectively (P<0.05).

**Table 5. Who is eligible to be a recipient of UTx?.**

| sex | age | group | N | Women who do not have a uterus congenitally (e.g. Rokitansky syndrome) | A woman who has had the uterus removed due to malignant disease | A woman who has had the uterus removed due to benign disease (e.g. uterine fibroid, adenomyosis) | A woman who has lost uterine function after endometritis, surgery for miscarriage (Asherman's syndrome). | Men who have a desire to conceive (but there is no report of such cases, and there is no research using animals. It is unknown whether it is possible or not.) | others | indecisive |
|---|---|---|---|---|---|---|---|---|---|---|
| total | | | 1600 | 52.9% | 53.4% | 46.8% | 46.4% | 11.2% | 0.9% | 33.3% |
| male | 20–29 | A | 200 | 44.0% | 41.5% | 37.5% | 36.5% | 14.5% [D,H] | 2.5% | 35.5% [E,F] |
| | 30–39 | B | 200 | 50.5% | 50.0% | 44.0% | 43.0% | 16.0% [D,G,H] | 0.5% | 32.5% [F] |
| | 40–49 | C | 200 | 43.0% | 52.5% | 46.0% | 44.5% | 10.0% | 0.5% | 42.5% [B,E,F,H] |
| | 50–59 | D | 200 | 40.5% | 46.5% | 40.5% | 41.5% | 6.0% | 0.0% | 43.5% [B,E,F,H] |
| female | 20–29 | E | 200 | 61.0% [A,B,C,D] | 60.5% [A,B,D] | 52.5% [A,D] | 50.5% [A] | 15.0% [D,H] | 0.5% | 25.5% |
| | 30–39 | F | 200 | 69.5% [A,B,C,D,G] | 66.0% [A,B,C,D,G] | 57.5% [A,B,C,D,G] | 60.0% [A,B,C,D,G] | 14.0% [D,H] | 1.0% | 21.5% |
| | 40–49 | G | 200 | 54.5% [A,C,D] | 54.0% [A] | 44.5% | 44.5% | 9.0% | 1.0% | 36.0% [E,F] |
| | 50–59 | H | 200 | 60.5% [A,C,D] | 56.0% [A] | 51.5% [A,D] | 51.0% [A] | 5.0% | 1.5% | 29.0% |

A,B,C,D,E,F,G,H denotes significantly high compared to group A,B,C,D,E,F,G, or H respectively (P<0.05).

**Table 6. Who is eligible to be a donor of UTx?.**

| sex | age | group | N | Mother | Sister | Relatives other than mother and sisters | Friends | Anonymous third party | Female with gender identity disorder (genetically, physically female but mentally male) | deceased donor (brain death or cardiac death) | others | indecisive |
|---|---|---|---|---|---|---|---|---|---|---|---|---|
| total | | | 1600 | 39.4% | 37.9% | 15.4% | 5.9% | 17.9% | 19.1% | 27.6% | 0.7% | 34.6% |
| male | 20–29 | A | 200 | 39.5% | 37.5% | 16.0% [H] | 12.0% [E,F,G,H] | 18.0% | 20.0% | 23.5% | 2.0% | 34.5% [F] |
| | 30–39 | B | 200 | 39.5% | 40.0% | 22.5% [F,G,H] | 7.0% | 17.00% | 16.5% | 20.0% | 0.0% | 33.0% |
| | 40–49 | C | 200 | 34.5% | 35.5% | 16.0% [H] | 6.5% | 13.0% | 15.5% | 22.5% | 0.5% | 42.0% [E,F] |
| | 50–59 | D | 200 | 40.5% | 36.5% | 15.5% | 6.0% | 12.0% | 13.5% | 23.0% | 0.5% | 42.5% [E,F] |
| female | 20–29 | E | 200 | 43 | 38.0% | 19.0% [H] | 4.5% | 24.0% [C,D] | 23.5% [D] | 34.5% [A,B,C,D] | 0.0% | 28.5% |
| | 30–39 | F | 200 | 42.5 | 40.0% | 13.5% | 5.0% | 24.0% [C,D] | 27.5% [B,C,D,G] | 36.0% [A,B,C,D] | 1.0% | 24.5% |
| | 40–49 | G | 200 | 37.5 | 38.5% | 12.0% | 3.0% | 17.0% | 16.5% | 28.0% | 0.5% | 36.0% [F] |
| | 50–59 | H | 200 | 38 | 37.5% | 9.0% | 3.0% | 18.0% | 19.5% | 33.5% [A,B,C,D] | 1.0% | 36.0% [F] |

A,B,C,D,E,F,G,H denotes significantly high compared to group A,B,C,D,E,F,G, or H respectively (P<0.05).

## Personal opinions

To obtain personal opinions, we asked whether respondents would choose to receive UTx, gestational surrogacy, or adoption, assuming that they or their partners suffered from absolute uterine factor infertility. In response, 4.3%, 4.1%, and 8.2% of participants would opt for UTx, gestational surrogacy, and adoption, respectively (Table 7). The proportions of those who would opt for UTx, gestational surrogacy, and adoption if their spouse wished to were 30.1%, 27.8%, and 32.1%, respectively. Among all age groups, 1–7% of participants would opt for UTx. A higher proportion (approximately 10%) of females than males would opt for adoption. More males than females would opt for UTx, gestational surrogacy, and adoption if their spouses wished to. Furthermore, a higher proportion of participants who would opt for UTx, gestational surrogacy, and adoption and those would choose these options if their spouse wished to had infertility (Table 8) or knew much about UTx (Table 9). When asked to choose the most favorable option, 10.1%, 5.8%, and 14.3% of respondents chose UTx, gestational surrogacy, and adoption, respectively (S6 Table). Particularly, higher proportions of males (26.3%) or females (16.1%) with infertility would choose UTx than those without infertility (Table 10). In addition, 25.5% of males and 37.5% of females who knew much about UTx selected it as a more favorable option (Table 11).

## Opinions about whether respondents would want to be donors for their daughters

To investigate the opinions of the participants from the donor's point of view, we asked whether the responders would like to be donors or whether they would like to ask their spouse to be a donor if their daughter was suffering from absolute uterine factor infertility. Among female participants, 15.9%, 16.4%, 7.1%, 19.5%, and 41.1%, responded "...want to be a

**Table 7. Attitudes towards each UTx, gestational surrogacy, and adoption assuming that the respondents or their spouse are suffering from absolute.**

| sex | age | group | N | Utx want to use | Utx want to use, if my spouse wishes | Utx not want to use, even if my spouse wishes | Utx indecisive | gestational surrogacy want to use | gestational surrogacy want to use, if my spouse wishes | gestational surrogacy not want to use, even if my spouse wishes | gestational surrogacy indecisive | adoption want to use | adoption want to use, if my spouse wishes | adoption not want to use, even if my spouse wishes | adoption indecisive |
|---|---|---|---|---|---|---|---|---|---|---|---|---|---|---|---|
| | | Total | 1600 | 4.3% | 30.1% | 24.8% | 40.9% | 4.1% | 27.8% | 29.6% | 38.6% | 8.2% | 32.1% | 18.6% | 41.1% |
| male | 20–29 | A | 200 | 6.0% H | 40.5% E,F,G,H | 13.5% | 40.0% | 4.5% | 39.0% E,F,G,H | 16.5% | 40.0% | 10.5% | 41.0% E,F,G,H | 13.0% | 35.5% |
| | 30–39 | B | 200 | 5.5% | 43.0% E,F,G,H | 17.0% | 34.5% | 5.0% H | 39.5% E,F,G,H | 21.5% | 34.0% | 7.5% | 41.5% E,F,G,H | 12.5% | 38.5% |
| | 40–49 | C | 200 | 2.5% | 47.5% D,E,F,G,H | 12.5% | 37.5% | 3.0% | 42.0% D,E,F,G,H | 21.5% | 33.5% | 2.5% | 41.0% E,F,G,H | 16.0% | 40.5% |
| | 50–59 | D | 200 | 2.0% | 34.0% F,G,H | 18.5% | 45.5% B | 2.5% | 30.0% F,G,H | 24.5% | 43.0% | 4.5% | 32.0% G | 18.0% | 45.5% |
| female | 20–29 | E | 200 | 6.0% H | 27.0% G,H | 23.0% A,C | 44.0% | 6.5% H | 25.0% G,H | 29.0% A | 39.5% | 12.0% C,D | 30.0% G | 17.0% | 41.0% |
| | 30–39 | F | 200 | 5.5% | 19.5% | 31.0% A,B,C,D | 44.0% | 5.5% H | 19.5% | 37.5% A,B,C,D | 37.5% | 12.0% C,D | 28.0% | 22.5% A,B | 37.5% |
| | 40–49 | G | 200 | 5.0% | 17.0% | 39.0% A,B,C,D,E | 39.0% | 5.0% H | 14.5% | 39.5% A,B,C,D,E | 41.0% | 9.0% C | 20.5% | 24.0% A,B | 46.5% A |
| | 50–59 | H | 200 | 1.5% | 12.5% | 43.5% A,B,C,D,E,F | 42.5% | 1.0% | 12.5% | 46.5% A,B,C,D,E | 40.0% | 7.5% C | 23.0% | 26.0% A,B,C,E | 43.5% |

A,B,C,D,E,F,G,H denotes significantly high compared to group A,B,C,D,E,F,G, or H respectively (P<0.05).

**Table 8. Attitudes towards UTx, gestational surrogacy, and adoption, assuming that the respondents or their spouses are suffering from absolute uterine factor infertility.**

| sex | infertility | group | N | UTx | | | | gestational surrogacy | | | | adoption | | | |
|---|---|---|---|---|---|---|---|---|---|---|---|---|---|---|---|
| | | | | want to use | want to use, if my spouse wishes | not want to use, even if my spouse wishes | indecisive | want to use | want to use, if my spouse wishes | not want to use, even if my spouse wishes | indecisive | want to use | want to use, if my spouse wishes | not want to use, even if my spouse wishes | indecisive |
| | | total | 1600 | 4.3% | 30.1% | 24.8% | 40.9% | 4.1% | 27.8% | 29.6% | 38.6% | 8.2% | 32.1% | 18.6% | 41.1% |
| male | + | A | 114 | 11.4% B,D | 57.0% B,C,D | 14.0% | 17.5% | 13.2% B,D | 49.1% B,C,D | 21.9% | 15.8% | 14.9% B,D | 53.5% B,C,D | 13.2% | 18.4% |
| | - | B | 686 | 2.8% | 38.6% C,D | 15.6% | 43.0% A,C | 2.2% | 35.7% C,D | 20.8% | 41.3% A,C | 4.8% | 36.4% D | 15.2% | 43.6% A,C |
| female | + | C | 193 | 10.4% B,D | 24.4% D | 31.1% A,B | 34.2% A | 8.8% B,D | 26.9% D | 34.7% A,B | 29.5% A | 16.1% B,D | 32.1% D | 19.2% | 32.6% A |
| | - | D | 607 | 2.6% | 17.3% | 35.1% A,B | 45.0% A,C | 3.1% | 15.0% | 39.2% A,B | 42.7% A,C | 8.2% B | 23.2% | 23.4% A,B | 45.1% A,C |

A,B,C,D denotes significantly high compared to group A,B,C, or D respectively (P<0.05).

donor," ". . .want to be a donor if there is no other donor," "UTx is acceptable, but I do not want to be a donor," ". . .do not want my daughter to undergo UTx," or "indecisive," respectively (Table 12). More participants aged 30–39 years than those aged 50–59 years, and more patients who knew much about UTx than those who had not heard about UTx would "want to be a donor." The proportion of participants who "do not want their daughter to undergo UTx" was highest in the 50–59 year age group.

Among males, 7.6%, 29.1%, 5.9%, 12.5%, and 44.9% responded ". . .would ask my spouse to be a donor," "would ask my spouse to be a donor if there was no other donor," "UTx is acceptable, but I do not want my spouse to be a donor," ". . .do not want my daughter to undergo UTx" or "indecisive," respectively. More participants with infertility than those without infertility, and more respondents those who knew much about UTx than those did not know much, responded that they would "want to ask my spouse to be a donor" (S7 Table, S8 Table). The proportion of participants who "do not want their daughter to undergo UTx" was highest in the 50–59 year age group.

## Discussion

We conducted a web-based survey on the attitudes of male and female members of the public aged 20–60 years toward UTx, gestational surrogacy, and adoption. To the best of our knowledge, this is the first large-scale survey on the public attitudes of the Japanese people including females older than 40 years and males. First, 36.5% and 31.0% of respondents agreed that UTx or gestational surrogacy should be approved, respectively. The frequency of respondents who demonstrated a positive attitude toward UTx or gestational surrogacy was higher among males and females with infertility than among those without infertility, and also higher among those who knew much about UTx than among those who had never heard about UTx. Second, the respondents would potentially choose to receive UTx (34.4%), gestational surrogacy (31.9%), and adoption (40.3%) if they or their partner suffered from absolute uterine factor infertility. Third, 10.1%, 5.8%, and 14.3% of participants chose UTx, gestational surrogacy, or adoption as the most favorable option, respectively. Particularly, the proportions of those who chose UTx was higher among males (26.3%) or females (16.1%) with infertility than among those without infertility, and also higher in the group of males (25.5%) or females (37.5%) who

**Table 9. Attitudes towards each UTx, gestational surrogacy, and adoption assuming that the respondents or their spouse are suffering from absolute uterine infertility.**

| sex | response | group | N | Utx | | | | gestational surrogacy | | | | adoption | | | |
|---|---|---|---|---|---|---|---|---|---|---|---|---|---|---|---|
| | | | | want to use | want to use, if my spouse wishes | not want to use, even if my spouse wishes | indecisive | want to use | want to use, if my spouse wishes | not want to use, even if my spouse wishes | indecisive | want to use | want to use, if my spouse wishes | not want to use, even if my spouse wishes | indecisive |
| | Total | | 1600 | 4.3% | 30.1% | 24.8% | 40.9% | 4.1% | 27.8% | 29.6% | 38.6% | 8.2% | 32.1% | 18.6% | 41.1% |
| male | Know very well | A | 47 | 31.9% B,C,D,F,G,H | 31.9% H | 21.3% B | 14.9% | 27.7% B,C,D,G,H | 36.2% H | 21.3% | 14.9% | 34.0% B,C,D,G,H | 31.9% | 19.1% | 14.9% |
| | Know well | B | 83 | 6.0% | 65.1% A,C,D,E,G,H | 4.8% | 24.1% | 4.8% D | 60.2% A,C,D,E,G,H | 14.5% | 20.5% | 8.4% | 57.8% A,C,D,G,H | 12.0% | 21.7% |
| | Have heard of it, but do not know much. | C | 217 | 3.2% | 43.3% G,H | 18.4% B | 35.0% A,E | 3.7% D | 38.7% G,H | 23.5% | 34.1% A,B | 5.5% | 41.0% G,H | 15.7% | 37.8% A,B,E,F |
| | Have never heard of it. | D | 453 | 1.1% | 36.9% G,H | 15.2% B | 46.8% A,B,C,E,F | 1.1% | 33.1% G,H | 21.0% | 44.8% A,B,C,E,F,G | 3.3% | 35.1% G,H | 14.6% | 47.0% A,B,C,E,F,G |
| female | Know very well | E | 32 | 37.5% B,C,D,F,G,H | 34.4% H | 12.5% | 15.6% | 31.3% B,C,D,F,G,H | 34.4% H | 15.6% | 18.8% | 34.4% B,C,D,G,H | 43.8% H | 9.4% | 12.5% |
| | Know well | F | 47 | 6.4% D | 61.7% A,C,D,E,G,H | 10.6% | 21.3% | 10.6% D,H | 46.8% G,H | 14.9% | 27.7% | 14.9% D | 55.3% A,D,G,H | 8.5% | 21.3% |
| | Have heard of it, but do not know much. | G | 187 | 3.7% D | 20.3% H | 35.3% B,C,D,E,F | 40.6% A,B,E,F | 3.7% D | 21.4% H | 39.6% A,B,C,D,E,F | 35.3% A,B | 10.7% D | 25.7% | 29.9% B,C,D,E,F,H | 33.7% A,E |
| | Have never heard of it. | H | 534 | 2.6% | 13.9% | 37.1% A,B,C,D,E,F | 46.4% A,B,C,E,F | 2.6% | 13.1% | 41.0% A,B,C,D,E,F | 43.3% A,B,C,E,F | 8.1% D | 21.5% | 21.7% B,D,F | 48.7% A,B,C,E,F,G |

A,B,C,D,E,F,G,H denotes significantly high compared to group A,B,C,D,E,F,G, or H respectively (P<0.05).

**Table 10. Attitudes towards each UTx, gestational surrogacy, and adoption assuming that the respondents or their spouse are suffering from absolute uterine factor infertility.**

| sex | infertility | group | N | Utx | gestational surrogacy | adoption | Do not want any way | indecisive |
|-----|-------------|-------|---|-----|----------------------|----------|---------------------|------------|
| | total | | 800 | 10.1% | 5.8% | 14.3% | 21.5% | 48.3% |
| male | + | A | 114 | 26.3% [B,C,D] | 14.0% [B,D] | 18.4% [B] | 11.4% | 29.8% |
| | - | B | 686 | 9.2% | 6.9% [D] | 11.7% | 16.6% | 55.7% [A,C,D] |
| female | + | C | 193 | 16.1% [B,D] | 8.3% [D] | 18.1% [B] | 20.2% | 37.3% |
| | - | D | 607 | 6.3% | 2.3% | 15.3% | 29.3% [A,B,C] | 46.8% [A,C] |

[A,B,C,D,E,F,G,H] denotes significantly high compared to group A,B,C,D,E,F,G, or H respectively (P<0.05).

knew much about UTx than among those who did not know much. Fourth, 32.3% of female respondents would want to be donors and 36.7% of male respondents would ask their wives to be donors if their daughters were suffering from absolute uterine factor infertility. Accordingly, UTx was the most acceptable option among UTx, gestational surrogacy, and adoption. The effects of gender, age, infertility, and level of knowledge regarding UTx are important in understanding the attitude toward UTx.

The present survey conducted among males and females showed that 36.5% of respondents were supportive of UTx for uterine factor infertility, and 17.0% were not. These results were influenced by gender, age, presence of infertility, or the level of knowledge regarding UTx. This suggests that the public has a positive attitude towards UTx in Japan. According to a previous report, 78% of respondents were supportive and 4% were against it in United States [11]. Also, in a previous survey of Japanese women younger than 40 years, 44.2% were supportive and 8.3% were against it [9]. The proportion found in our study appears to be lower than those reported in previous studies; however, this could be due to the reduction in the number of participants older than 40 years who were supportive of UTx. In fact, among women in their 30s, 45% were supportive, and 15.5% were against it, which corroborates our previous finding in Japan [10].

**Table 11. Attitudes towards each UTx, gestational surrogacy, and adoption assuming that the respondents or their spouse are suffer from absolute uterine factor infertility.**

| sex | age | group | N | Utx | gestational surrogacy | adoption | Do not want any way | indecisive |
|-----|-----|-------|---|-----|----------------------|----------|---------------------|------------|
| | total | | 800 | 10.1% | 5.8% | 14.3% | 21.5% | 48.3% |
| male | Know very well | A | 47 | 25.5% [C,D,G,H] | 10.6% [H] | 21.3% | 21.3% | 21.3% |
| | Know well | B | 83 | 19.3% [D,G,H] | 19.3% [C,D,G,H] | 19.3% [D] | 12.0% | 30.1% |
| | Have heard of it, but do not know much. | C | 217 | 12.9% [H] | 6.9% [H] | 12.4% | 15.7% | 52.1% [A,B,E,F,G] |
| | Have never heard of it. | D | 453 | 8.2% | 6.0% [H] | 10.6% | 16.1% | 59.2% [A,B,E,F,G,H] |
| female | Know very well | E | 32 | 37.5% [C,D,G,H] | 15.6% [G,H] | 15.6% | 6.3% | 25.0% |
| | Know well | F | 47 | 25.5% [C,D,G,H] | 10.6% [H] | 23.4% [D] | 12.8% | 27.7% |
| | Have heard of it, but do not know much. | G | 187 | 9.1% | 4.3% | 16.6% [D] | 31.0% [B,C,D,E,F] | 39.0% [A] |
| | Have never heard of it. | H | 534 | 5.2% | 2.2% | 15.2% [D] | 28.3% [B,C,D,E,F] | 49.1% [A,B,E,F,G] |

[A,B,C,D,E,F,G,H] denotes significantly high compared to group A,B,C,D,E,F,G, or H respectively (P<0.05).

**Table 12. Do you want to be a donor for your daughter with absolute uterine factor infertility.**

| sex | age | group | N | Do you want to be a donor for your daughter with absolute uterine factor infertility? | | | | |
|---|---|---|---|---|---|---|---|---|
| | | | | Want to be a donor for our daughter | Want to be a donor if there are no other donors | Uterine transplantation is acceptable, but I do not want to be a donor. | Do not want our daughter to undergo UTx, no matter who the donor is. | indecisive |
| | total | | 800 | 15.9% | 16.4% | 7.1% | 19.5% | 41.1% |
| female | 20–29 | E | 200 | 15.5% | 20.5% | 9.5% | 15.5% | 39.0% |
| | 30–39 | F | 200 | 21.5% [H] | 16.5% | 8.0% | 13.0% | 41.0% |
| | 40–49 | G | 200 | 16.5% | 14.0% | 5.0% | 20.5% | 44.0% |
| | 50–59 | H | 200 | 10.0% | 14.5% | 6.0% | 29.0% [E,F] | 40.5% |
| | | | | Do you ask your spouse to be a donor for your daughter with absolute uterine factor? | | | | |
| | | | | Want to ask my spouse to be a donor for our daughter | Want to ask my spouse to be a donor if there are no other donors | Uterine transplantation is acceptable, but I do not want my spouse to be a donor. | Do not want our daughter to undergo UTx, no matter who the donor is. | indecisive |
| | total | | 800 | 7.6% | 29.1% | 5.9% | 12.5% | 44.9% |
| male | 20–29 | A | 200 | 7.5% | 28.0% | 7.0% | 13.5% | 44.0% |
| | 30–39 | B | 200 | 11.0% | 31.5% | 7.0% | 10.0% | 40.5% |
| | 40–49 | C | 200 | 6.5% | 37.0% [B] | 3.0% | 8.0% | 45.5% |
| | 50–59 | D | 200 | 5.5% | 20.0% | 6.5% | 18.5% [B,C] | 31.1% |

[A,B,C,D,E,F,G,H] denotes significantly high compared to group A,B,C,D,E,F,G, or H respectively (P<0.05).

In our study, 31.5% of participants supported gestational surrogacy, while 19.9% did not. In our previous survey, 40.9% of respondents favored gestational surrogacy, while 21.8% did not [10]. By comparison, the proportion of participants who had a positive attitude toward gestational surrogacy in this survey seemed to be lower. This is because UTx is originally an alternative to gestational surrogacy, and the option of UTx may have an impact on the attitude toward gestational surrogacy, probably due to the difference in contexts when comparing gestational surrogacy with UTx.

The main reasons for social approval of UTx were "UTx will give hope to patients with uterine infertility" and "UTx enables patients to be genetic and birth parents." This is similar to the previous report [9]. On the other hand, the top reasons why participants felt that UTx should not be approved were "the risk of surgery itself is high," "pregnancy and childbirth should be natural," "the uterus is not a vital organ," and "the risk of pregnancy and delivery is high." This trend was also in line with the previous report [9]. UTx has gained recognition as the latest technology that allows patients with uterine factor infertility to become pregnant and experience childbirth. On the other hand, because UTx is still at the stage of clinical research, there is much concern about its safety. Donor surgery is a major surgery that lasts 10 to 13 hours [12]. For this reason, we believe that follow-up and reporting of the prognosis and safety of donors, recipients, and resulting children should be continued. In that regard, the report of having obtained a living child by UTx from a deceased donor seems to have presented an important

option for decreasing the risks associated with donor surgery [3]. This would also avoid the ethical issues with donors.

Assuming that they or their partner were suffering from uterine factor infertility, 34.4%, 31.9%, and 40.3% of respondents would opt for UTx, gestational surrogacy, and adoption, respectively, which showed no clear difference among them. Also, compared with females, males were significantly more likely to opt for UTx and gestational surrogacy. The same trend was obtained previously when we assessed the attitude of the public toward surrogate pregnancy [10]. This might be due to men not being physically involved during the procedure and having a strong desire for offspring. In addition, a higher proportion of participants with infertility would opt for either method. The fact that the experience of infertility affects attitudes toward UTx and surrogacy is understandable, considering that infertility has a major impact on psychological well-being and sexuality [13, 14].

In the single choice question, 14.3% opted for adoption, 10.1% for UTx, 5.8% for surrogate conception, 21.5% did not like either method, and 48.3% were indecisive. Among participants with infertility and those who knew about uterine transplantation, the number of indecisive participants was lower, and a higher number chose adoption, UTx, or gestational surrogacy. Particularly, among women who were familiar with UTx, the proportion of those who opted for UTx was 37.5%, which far exceeds the 15.6% obtained for surrogate pregnancy and adoption. Not only is it important to raise awareness about UTx, but the results also suggested that gaining knowledge about UTx may increase the number of supportive opinions. Therefore, it may be important to provide more information on UTx and discuss it.

Further, we surveyed the attitudes of participants toward being a donor of UTx. Approximately 30% of the respondents would want to be donors or would ask their wives to be donors if their daughter suffered from absolute uterine factor infertility. In addition, this frequency was higher among infertile participants or those who knew about UTx. Therefore, mothers are more likely to be donors if UTx is approved in Japan. However, this trend may change in the future if UTx from deceased donors becomes widespread.

In many previous papers, UTx has been described as the only option to have a genetic link to the child when gestational surrogacy is prohibited. In addition, even when live donor UTx is performed, UTx is considered superior to surrogacy, since it is supposed to be a less morally problematic alternative to gestational surrogacy [15]. However, some authors are skeptical of the assumption that UTx is morally superior to gestational surrogacy [15]. In addition, it has been reported that the perinatal risk during gestational surrogacy is not significantly different from that attributable to *in vitro* fertilization using fresh embryos [16, 17]. There have been reports of preeclampsia or preterm birth after UTx [1], and it is possible that the risk may surpass that of gestational surrogacy regarding the perinatal outcome after UTx, including the use of immunosuppressants. In Japan, although the guidelines of the Japan Society of Obstetrics and Gynecology prohibit surrogacy, no legislation has been enacted in this regard [7, 8]. In this survey, more than 30% of respondents had supportive opinions about UTx or gestational surrogacy. Therefore, it is necessary to discuss the ethical issues regarding gestational surrogacy in parallel with those regarding UTx.

The present study has some limitations. First, this study was a cross-sectional study and cannot be used to explore a causal relationship. Second, there is the possibility of survey selection bias and issues of generalizability. Third, we did not assess the level of knowledge and understanding about UTx or gestational surrogacy among the respondents. Despite these limitations, by conducting a web-based questionnaire survey of the general population, we obtained a large sample that included females older than 40 years and males. Therefore, the findings of this study reflect the public attitudes toward UTx and gestational surrogacy in Japan.

## Conclusions

In the present study, we clarified the attitudes of male and female members of the public toward UTx, surrogacy, and adoption in Japan. Our results suggested that UTx is the most acceptable option among UTx, gestational surrogacy, and adoption. The effects of gender, age, infertility, and the level of understanding of UTx affected the attitude toward UTx. On the other hand, there are concerns about the safety of UTx for recipients, donors, and babies. It is important to continue to understand public attitudes to inform the development and safety of UTx, to enhance the discussion on the ethical consensus regarding UTx.

## Supporting information

**S1 Fig. Reference materials to deepen the understanding of respondents.**
(PPTX)

**S1 Document. The questionnaire used in this study.**
(DOCX)

**S1 Table. The demographic data of each group respondents (N = 1600).**
(XLSX)

**S2 Table. Why do you think that we should approve UTx?.**
(XLSX)

**S3 Table. Why do you think that UTx should not be approved? (N = 272).**
(XLSX)

**S4 Table. Why do you think that we should approve a surrogate pregnancy using the womb of a third party?.**
(XLSX)

**S5 Table. Why do you think that gestational surrogacy should not be approved? (N = 318).**
(XLSX)

**S6 Table. Attitudes toward each UTx, gestational surrogacy, and adoption assuming that the respondents or their spouse present with absolute uterine factor infertility.**
(XLSX)

**S7 Table. Do you want to be a donor for your daughter with absolute uterine factor infertility?.**
(XLSX)

**S8 Table. Do you want to be a donor for your daughter with absolute uterine factor infertility?.**
(XLSX)

## Acknowledgments

We would like to thank Editage (www.editage.jp) for the English language review.

## Author Contributions

**Conceptualization:** Akari Nakazawa, Miyuki Harada, Osamu Wada-Hiraike, Tomoyuki Fujii, Minoru Irahara, Yutaka Osuga.

**Data curation:** Akari Nakazawa, Tetsuya Hirata, Tomoko Arakawa, Natsuki Nagashima, Shi-nya Fukuda, Kazuaki Neriishi, Yasushi Hirota, Kaori Koga, Osamu Wada-Hiraike.

**Formal analysis:** Tetsuya Hirata.

**Funding acquisition:** Tetsuya Hirata, Minoru Irahara, Yutaka Osuga.

**Investigation:** Tetsuya Hirata, Shinya Fukuda, Kaori Koga.

**Methodology:** Akari Nakazawa, Tetsuya Hirata, Osamu Wada-Hiraike, Yoshio Koizumi, Tomoyuki Fujii.

**Project administration:** Tetsuya Hirata.

**Supervision:** Yutaka Osuga.

**Validation:** Tetsuya Hirata, Miyuki Harada, Yasushi Hirota, Osamu Wada-Hiraike, Yoshio Koizumi, Tomoyuki Fujii, Minoru Irahara, Yutaka Osuga.

**Writing – original draft:** Akari Nakazawa, Tetsuya Hirata.

**Writing – review & editing:** Akari Nakazawa, Tetsuya Hirata.

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
