## [Decision Letter · Decision Letter 0]

10 Sep 2019

[EXSCINDED]

PONE-D-19-20468

A survey of public attitudes toward uterus transplantation, surrogacy, and adoption in Japan

PLOS ONE

Dear Dr. Hirata,

Thank you for submitting your manuscript to PLOS ONE. After careful consideration, we feel that it has merit but does not fully meet PLOS ONE’s publication criteria as it currently stands. Therefore, we invite you to submit a revised version of the manuscript that addresses the points raised during the review process.

The manuscript and the reviewers’ comments were carefully evaluated. The manuscript is a well written manuscript with clear methodology. It was appreciated by the Reviewers, that suggested only some minor revision, that are in detail reported in the Reviewers’ comments.

We would appreciate receiving your revised manuscript by Oct 25 2019 11:59PM. To enhance the reproducibility of your results, we recommend that if applicable you deposit your laboratory protocols in protocols.io, where a protocol can be assigned its own identifier (DOI) such that it can be cited independently in the future. For instructions see: http://journals.plos.org/plosone/s/submission-guidelines#loc-laboratory-protocols

We look forward to receiving your revised manuscript.

Kind regards,

Simone Garzon

Academic Editor

PLOS ONE

Journal Requirements:

2. Please provide additional details regarding participant consent. In the ethics statement in the Methods and online submission information, please ensure that you have specified (1) whether consent was informed and (2) what type you obtained (for instance, written or verbal). If your study included minors, state whether you obtained consent from parents or guardians. If the need for consent was waived by the ethics committee, please include this information.

We additionally ask that you please include in your methods section, the name of the company which conducted the survey for this study.

"This work was supported by grants from the Ministry of Education, Culture, Sports, Science and Technology and the Japan Agency for Medical Research and Development. We would like to thank Editage (www.editage.jp) for the English language review."

"No"

Reviewers' comments:

Reviewer's Responses to Questions

**Comments to the Author**

1. Is the manuscript technically sound, and do the data support the conclusions?

Reviewer #1: Yes

Reviewer #2: Partly

2. Has the statistical analysis been performed appropriately and rigorously? 

Reviewer #1: Yes

Reviewer #2: Yes

3. Have the authors made all data underlying the findings in their manuscript fully available?

Reviewer #1: Yes

Reviewer #2: Yes

4. Is the manuscript presented in an intelligible fashion and written in standard English?

Reviewer #1: Yes

Reviewer #2: No

5. Review Comments to the Author

Reviewer #1: Very good paper. The authors addressed the pros and cons of all alternative methods in a good and clear way.

Minor change: Line 309, page 25. UTx should be changed to Surrogacy

Reviewer #2: I read with great interest the Manuscript titled “A survey of public attitudes toward uterus transplantation, surrogacy, and adoption in Japan” (PONE-D-19-20468), which falls within the aim of PLOS ONE. In my honest opinion, the topic is interesting enough to attract the readers’ attention. Methodology is accurate and conclusions are supported by the data analysis. Nevertheless, authors should clarify some points and improve the discussion citing relevant and novel key articles about the topic.

Authors should consider the following recommendations:

- Manuscript should be further revised by a native English speaker to improve its readability in some points.

- Methods. I would suggest reporting how the minimum of 200 samples for each group was calculated.

- Methods. I would suggest reporting in the methods section which was the definition of “infertility” for the subjects. Moreover, was the definition of “experience of infertility” confirmed by the subsequent evaluation of the reported infertility treatment?

- Methods. I would suggest better explaining why the maximum age was 60 years, and older subjects were excluded by the Authors.

- I could not find any information regarding the informed consent of enrolled patients. Did author obtain informed consent for each patient? Conversely, this point may raise serious concern from the ethical point of view.

- I suggest discussing, at least briefly, how the condition of infertility may have a significant and detrimental impact of psychological well-being and sexuality (refer to: PMID: 27750491; PMID: 29336414; PMID: 31466493).

6. PLOS authors have the option to publish the peer review history of their article (what does this mean?). If published, this will include your full peer review and any attached files.

Reviewer #1: Yes: randa akouri

Reviewer #2: No

---

## [Author Response · Author response to Decision Letter 0]

21 Sep 2019

Sep 19, 2019

Editorial Board

PLOS ONE

Dear Editor

　We really appreciated editors and reviewers for the important suggestions, which improved our manuscript. We are also very pleased that editors and reviewers acknowledged the issue on Japanese attitudes towards uterus transplantation, gestational surrogacy, and adoption. We believe that these survey data will help to reach a national consensus on that.

Please find the revised version of our manuscript entitled “A survey of public attitudes towards uterus transplantation, surrogacy, and adoption in Japan.” In the attached manuscript, we described the changes made in response to the reviewer’s comments point-by-point as follows. 

Reviewer #1

We appreciated the reviewer for the comments.

Comment #1

 UTx should be changed to Surrogacy in line 309. 

Response

 We thank the reviewer. This was typing error, which may confuse the readers. 

We corrected it as follows.

 “However, some authors are skeptical of the assumption that UTx is morally superior to gestational surrogacy.”

Reviewer #2

We appreciated reviewers for the critical comments and important suggestions that have helped us to improve our manuscript.

Comment #1

 The manuscript should be revised by a native English speaker. 

Response 

 Thank you for your suggestion. Our manuscript was subjected to English editing company, and was revised by them. All occurrences of “towards” have been changed to “toward” for consistency. 

Comment #2

 The authors should describe how to isolate 200 samples from each group.

Response 

 We appreciated the reviewer for the important comment. Our explanation may have been confusing. We asked the internet research company for the collection of samples. The upper limit of each sample collection was determined to be 250 samples. They closed the collection, when the number of samples reached 250 samples in each group. Among them, 200 samples were selected randomly. Through this process, the internet research company extracted 1600 samples with equal segment of age and gender. After that, we received the results of extracted 1600 samples, and subsequently used them for the analysis. 

 We change “planned to collect….” to “asked the internet research company to collect ….” in line 93, as follows.

 ‘We asked the internet research company to collect 200 samples for each group of men and women in their 20's, 30’s, 40’s, and 50's.’

Comment #3

 The authors should describe the definition of “infertility”. 

Response 

 We appreciated reviewers for the important comment. In Japan, “infertility” is defined as unsuccessful attempted conception for 12 months or longer. We added “In Japan, infertility is defined as unsuccessful attempted conception for 12 months or longer.” in line 87.

Comment #4

 Was the definition of “experience of infertility” confirmed by the subsequent evaluation of the reported infertility treatment? 

Response 

 We appreciated reviewers for the comments. In this study, we asked whether respondents have any experience of infertility. The subjects were those who suffered from infertility even if they had not undergone fertility treatment or screening. In this study, of the 307 respondents with experience of infertility, 93 did not undergo infertility screening or treatment. We appreciated reviewers for the important comments. We added it in the result section in line 114, as follows.

 “In this study, we asked whether respondents have any experience of infertility. The subjects were those who had infertility even if they had not undergone fertility treatment or screening. In this study, of the 307 respondents with experience of infertility, 93 did not undergo infertility screening or treatments.”

Comment #5

 The authors should explain the reason why the maximum age was 60 years.

Response 

　Because we assumed that uterus transplantation donors and recipients could be women aged 20 to under 60, this study targeted men and women between the ages of 20 and 60. We appreciated the reviewer for the important comment. We added it in line 91, as follows. 

“Because we assumed that uterus transplantation donors and recipients could be women aged 20 to under 60, this study targeted men and women between the ages of 20 and 60.”

Comment #6

 The authors should give the information regarding the informed consent.

Response 

 We appreciated the reviewer for the important suggestion. We obtained informed consent before respondents answered questionnaire. Respondents read the study description and then choose to participate or not to participate in the study. Respondents started to answer the questionnaire when they selected “Yes” to a question of whether to participate in the survey. We also explained that respondents can stop answering if they want to stop the survey halfway. We added it in line 79, as follows. 

“They provided informed consent before answering the questionnaire. Respondents read the study description and then chose to participate or not to participate in the study. Respondents started to answer the questionnaire after they selected “Yes” to a question of whether to participate in the survey. We also explained that respondents can stop answering if they want to stop the questionnaire halfway.”

Comment #7

 The authors should discuss how the condition of infertility may have a significant impact of psychological well-being and sexuality.

Response 

 We appreciated reviewers for the suggestion. We discuss about it in line 301, as follows.

“The fact that the experience of infertility affects attitudes towards UTx and surrogacy is understandable, considering that infertility has a major impact on psychological well-being and sexuality.”

　Journal Requirements

#1 The authors should describe the participant consent.

 Response 

We obtained informed consent before respondents answered questionnaire. Respondents read the study description and then choose to participate or not to participate in the study. Respondents started to answer the questionnaire when they selected “Yes” to a question of whether to participate in the survey. We also explained that respondents can stop answering and withdraw all answers if they want to withdraw from our survey halfway. We added it in line 79, as follows. 

“They provided informed consent before answering the questionnaire. Respondents read the study description and then chose to participate or not to participate in the study. Respondents started to answer the questionnaire after they selected “Yes” to a question of whether to participate in the survey. We also explained that respondents can stop answering and withdraw all answers if they want to withdraw from our survey halfway.”

#2 The authors should state how to obtain the informed consent, if the survey includes minors.

Response

Our study does not include minors. We added it in line 92, as follows.

“Because we assumed that uterus transplantation donors and recipients could be women aged 20 to under 60, this study targeted men and women between the ages of 20 and 60.”

#3 The authors should add the name of internet research company in the method section.

Response

 We added the name of company in line 77. The name is NEO MARKETING INC.

#4 The authors should remove the funding information from “Acknowledgments”.

Response 

 We removed the funding statement from Acknowledgments. We would like to add funding information in Funding Statement of online submission form, as follows. 

Grant from the Ministry of Education, Culture, Sports, Science and Technology 

Grant from the Japan Agency for Medical Research and Development.

We hope that the revised that the revised version of our paper is now suitable for publication in PLOS ONE and look forward to hearing from you. 

 Sincerely.

 Tetsuya Hirata, M.D., Ph.D.

---

## [Editor Report · Decision Letter 1]

25 Sep 2019

A survey of public attitudes toward uterus transplantation, surrogacy, and adoption in Japan

PONE-D-19-20468R1

Dear Dr. Hirata,

We are pleased to inform you that your manuscript has been judged scientifically suitable for publication and will be formally accepted for publication once it complies with all outstanding technical requirements.

With kind regards,

Simone Garzon

Academic Editor

PLOS ONE

Additional Editor Comments (optional):

The manuscript is well written with clear methodology. It was appreciated by the Reviewers, who suggested only some minor revision completely addressed by the Authors.
---

## [Editor Report · Acceptance letter]

30 Sep 2019

PONE-D-19-20468R1 

A survey of public attitudes toward uterus transplantation, surrogacy, and adoption in Japan 

Dear Dr. Hirata:

I am pleased to inform you that your manuscript has been deemed suitable for publication in PLOS ONE. Congratulations! Your manuscript is now with our production department. 

With kind regards,

on behalf of

Dr. Simone Garzon 

Academic Editor

PLOS ONE